# Nutraceutical Potential of *Lentinula edodes*’ Spent Mushroom Substrate: A Comprehensive Study on Phenolic Composition, Antioxidant Activity, and Antibacterial Effects

**DOI:** 10.3390/jof9121200

**Published:** 2023-12-15

**Authors:** Filipa Baptista, Joana Campos, Valéria Costa-Silva, Ana Rita Pinto, Maria José Saavedra, Luis Mendes Ferreira, Miguel Rodrigues, Ana Novo Barros

**Affiliations:** 1Centre for the Research and Technology of Agro-Environmental and Biological Sciences (CITAB), University de Trás-os-Montes e Alto Douro (UTAD), 5000-801 Vila Real, Portugal; joanacampos@utad.pt (J.C.); al60149@alunos.utad.pt (A.R.P.); saavedra@utad.pt (M.J.S.); lmf@utad.pt (L.M.F.); mrodrigu@utad.pt (M.R.); 2CECAV—Animal and Veterinary Research Centre, University of Trás-os-Montes and Alto Douro (UTAD), Quinta de Prados, 5000-801 Vila Real, Portugal; valeriasilva@utad.pt

**Keywords:** *Lentinula edodes*, spent mushroom substrate, extraction methods, nutraceuticals, prebiotic supplementation, valuable by-product resources, circular economy

## Abstract

*Lentinula edodes*, commonly known as shiitake mushroom, is renowned for its potential health advantages. This research delves into the often-overlooked by-product of shiitake cultivation, namely spent mushroom substrate (SMS), to explore its nutraceutical properties. The SMS samples were collected and subjected to different extraction methods, namely short or long agitation, and ultrasound-assisted extractions using different temperatures and distilled water or a 50% (*v*/*v*) ethanol as solvents. The extracts were tested for phenolic content (total phenols, *ortho*-diphenols, and flavonoids), antioxidant capacity (DPPH, 2,2-diphenyl-1 picrylhydrazyl; ABTS, 2,2’-azino-bis-3-ethylbenzothiazoline-6-sulfonic acid; and FRAP, ferric reducing antioxidant power), and antibacterial activity. The different extraction methods revealed substantial variations (*p* < 0.05) in phenolic composition and antioxidant capacity. The highest phenolic content and antioxidant capacity were achieved using 24 h extraction, agitation, 50 °C, and ethanol as the solvent. Furthermore, the extracted compounds displayed antibacterial activity in specific tested bacterial strains. This study highlights the nutraceutical potential of *L. edodes*’ SMS, positioning it as a valuable dietary supplement for animal nutrition, with emphasis on its prebiotic properties. Hence, this research unveils the promising health benefits of SMS in both human and animal nutrition.

## 1. Introduction

Currently, there are around 12,000 species of fungi that can be classified as mushrooms, including over 2000 edible and therapeutic mushroom species, many of which are widely consumed [1,2,3]. The five primary genera constitute around 85% of the world’s mushroom supply, with *Lentinula* being the principal genus contributing to about 26% of the world’s supply, followed by *Auricularia* spp. (21%), *Pleurotus ostreatus* (16%), *Agaricus bisporus* (11%), and *Flammulina velutipes* (7%) [4,5,6].

Remarkably, multifaceted benefits of the Shiitake mushroom have been reported due to its bioactive compounds, such as ergosterol, glucans, lectines, and polysaccharides. These compounds confer a wide array of properties, such as antimicrobial, antitumor, antiviral, anti-inflammatory, antioxidant properties, among others [7]. These findings are especially relevant in the context of the anticipated growth in mushroom production. This increase is driven by the growing recognition of mushrooms as a crucial cash crop and the increase demand for higher nutritional standards [8,9,10,11,12]. Particularly, in 2018, the global mushroom market was around 13 million tons, and it is estimated to reach 21 million tons by 2026 [4,8,13,14], representing an annual growth rate of 6.41% during the forecast period [4]. In the year 2013, China dominated the global mushroom production landscape, boasting an impressive 87% share and producing an astounding quantity exceeding 30 million tons of mushrooms [15]. The remainder of Asia produced about 1.3 million tons whereas other countries combined produced around 3.1 billion kg [4]. In Europe, the mushroom production reached 1 million tons in 2013, with a value of EUR 1.8 billion [16].

Nevertheless, the mushroom industry generates substantial waste, with spent mushroom substrate (SMS) emerging as the primary residue post-harvest, with an estimated production of around 5 kg per kg of fresh mushrooms harvested [8,9,17]. The SMS is typically regarded as agricultural waste, constituting a blend of residual fungal mycelia, along with a diverse range of lignocellulosic materials (e.g., saw-dust, corn cob, cottonseed hull, livestock litter, manure, straw, wood chips, enzymes), and is commonly disposed of through methods such as landfilling, incineration, or composting, often without proper utilization [8,16,17,18,19]. In fact, over the last decade, a total of over 60 million tons of SMS was generated due to the exponential expansion of the global edible mushroom industry [8,14]. Particularly in Europe, the current costs associated with SMS disposal range from EUR 10 to 50 per ton, entailing a substantial financial burden that can reach up to EUR 150 million annually for the mushroom industry [16].

Therefore, it becomes imperative to mitigate the environmental impact by devising suitable disposal strategies, thereby fostering the sustainable progress of the edible mushroom industry [20]. A central challenge lies in adopting the emerging circular economy concept to efficiently transform these by-products, when their useful life ends, into valuable resources. This approach ensures their seamless integration into the production chain, effectively reducing waste generation and environmental impact [8,12,15,21,22,23]. The European Union (EU) is actively endorsing this paradigm, aiming to create novel opportunities and innovative markets, fortify business resilience against resource scarcity, and enhance competitiveness [22,24].

Considerable efforts have been dedicated to formulating strategies for the efficient valorisation of SMS. As illustrated in Figure 1, SMS can be harnessed to create premium-quality composts [25,26], fertilizers and soil amendments, and vermicomposting [27,28]. An alternative approach is the recycling of the spent substrate, supplemented with additional nutrients, for new cultivation cycles of some mushroom species [8,9]. Furthermore, SMS can be used as feedstock for the production of biofuel (e.g., biogas, bioethanol, biohydrogen, volatile fatty acids) [8,20,29], and materials such as polymer foams that can be applied as packaging material or in construction. Simultaneously, a range of other potential applications has been put forward, encompassing diverse areas such as acoustic dampening, absorbent materials, textiles, paper products, as well as components for vehicles and electronics [9,30,31]. Arguably, one of the most relevant uses lies with enzymes and other bioactive compounds extraction from SMS [32,33] and its use as animal feed supplements [20,34]. Also, it is important to understand the potential of these by-products and to develop technologies and/or methodologies to exploit them as products offering additional nutritional or pharmaceutical advantages is equally significant [35].

Given the challenges encountered within the mushroom industry, the objective of this study was to explore the viability of repurposing residues from the industry, specifically *Lentinula edodes*’ SMS, as potential sources of nutraceutical or pharmaceutical natural compounds. The study involved quantifying the phenolic compounds within the SMS, evaluating its antioxidant capacity, and assessing its potential antibacterial activity. This assessment encompassed a comparative analysis of various extraction methods and solvents, both of which significantly influence the properties of the obtained extracts.

## 2. Materials and Methods

### 2.1. Chemical Products

The compounds gallic acid, sodium molybdate, sodium carbonate, sodium acetate, catechin potassium persulfate, methanol, 2,2′-Azino-bis (3-ethylbenzothiazoline-6-sulfonic acid) diammonium salt (ABTS^•+^), 6-Hidroxy-2,5,7,8-tetra-methylchromone-2-carboxylic acid (Trolox) and 2,2-diphenyl-1-picrylhydrazyl (DPPH^•^) were obtained from Sigma-Aldrich (Steinheim, Germany). Folin–Ciocalteu’s reagent, formic acid (pro-analysis), and acetonitrile (HPLC gradient grade) were purchased from Panreac (Panreac Química S.L.U., Barcelona, Spain). Ultrapure water was obtained using a Millipore water purification system. Additionally, aluminum chloride, sodium nitrite, and sodium hydroxide were purchased from Merck (Merck, Darmstadt, Germany). All material used, including culture media and antibiotics used for the antimicrobial activity assessment, were purchased from Oxoid (Thermo Fisher Scientific Inc., Lisbon, Portugal).

### 2.2. Sampling and Extraction Methods

The spent mushroom substrate samples (*L. edodes*) were collected from the FLORESTA. VIVA company (Amarante, Portugal). All the samples were dried at 50 °C and finely milled through a 0.5 mm mesh. For the extraction methods, 30 g of the previously dried spent mushroom substrate were added to 300 mL of a solvent (ratio 1:10); two different solvents were used, distilled water and a 50% (*v*/*v*) ethanol solution, as presented in Figure 2 and Table 1.

The first method applied was a short extraction, meaning the substrate was stirred with the solvent for 1 h. The second method was a longer version of the previous method (24 h) and two temperatures were tested to observe the effect of temperature in the extraction procedure, both methods performed according to Hu et al. [36] with some modifications. The third method was a 20 min ultrasound-assisted extraction (50 °C, 110 W), performed according to Agcam et al. [37] with some modifications, and the fourth and last method was a combination of the second and third methods. This resulted in a total of 12 extracts. An overview of the extraction methods previously explained can be found in Table 1. All obtained extracts were lyophilized and stored at −80 °C, until further analysis.

### 2.3. Determination of Phenolic Content

All the extracts were redissolved in distilled water in a concentration of 1 mg of extract per mL of distilled water, resulting in a total of three replicas of 10 mL of sample per extract. The analyses performed for polyphenolic determination were based on the content of total phenols, *ortho*-diphenols, and flavonoids. For all analysis, three replicates (n = 3) were performed in 96-well microplates (Frilabo, Milheirós, Portugal) and the absorbance measurements were performed with a Multiskan equipment (Thermo Fisher Scientific^®^, Vantaa, Finland).

The total polyphenols content of the extracts was determined using the Folin–Ciocalteu reagent, according to the procedure described by Yu et al. [38]. Briefly explained: 20 μL of gallic acid standard, sample or distilled water (blank), 100 μL of Folin–Ciocalteu previously diluted in water (1:10 H_2_O), and 80 μL of sodium carbonate (7.5%) were added to each well of the 96-well microplate. After a 30 min incubation period at 40–45 °C, the absorbance was measured at 725 nm and the values obtained were compared with a calibration curve of gallic acid (5–250 mg L^−1^). The results were expressed in milligram of gallic acid equivalent per gram of sample (mg GA g^−1^). This procedure consisted of the formation of a blue colored complex, originated by the reduction in phospholate-phosphomolybdate by phenolic compounds.

The evaluation of *ortho*-diphenols content consisted of the method presented by Yu et al. [38]. Briefly explained, similarly to the previous method, gallic acid was used as standard for the calibration curve (5–200 mg per gram of solution range). To that was added a 160 μL of sample, standard, or distilled water (blank), followed by 40 μL of sodium molybdate to each well of the microplate. After an incubation period of 15 min, the absorbance was measured at 375 nm. The content of *ortho*-diphenols was expressed as milligram of gallic acid per gram of sample (mg GA g^−1^).

The determination of the flavonoid content was based on the colorimetric assay described by Yu et al. [38]. Briefly explained, 24 μL of standard, sample, or distilled water (blank), and 28 μL of sodium nitrite (NaNO_2_) were added to the microplate. After a 5 min wait period, 28 μL of aluminum chloride (AlCl_3_) were added. After a 6 min wait period, 120 μL of sodium hydroxide (NaOH) were added. The microplate was then stirred for 30 s, and the absorbance was measured at 510 nm. The results were expressed as milligram of catechin per gram of sample (mg CAT g^−1^). For the calibration curve, catechin standards were used in a 5–200 mg per gram of solution range.

### 2.4. Determination of Antioxidant Capacity

The analyses performed for antioxidant capacity determination were DPPH, ABTS, and FRAP. For all analysis, three replicates (n = 3) were performed in 96-well microplates (Frilabo, Milheirós, Portugal) and the absorbance measurements were performed with resort to a Multiskan equipment (Thermo Fisher Scientific^®^, Vantaa, Finland).

The DPPH assay used in this work is based on the method outlined by Yu et al. [38]. A standard solution of 24 mg of DPPH in 100 mL ethanol was prepared (DPPH radical). The DPPH stock solution was prepared by diluting the DPPH standard solution in methanol until obtaining an absorbance of 0.93 ± 0.04 at 515 nm. Afterwards, 190 μL of this solution, and 10 μL of Trolox standard, sample, or distilled water (blank) were added to each well of the microplate. After a 30 min incubation period, the absorbance was measured at 520 nm. The results obtained were expressed in mmol of Trolox per gram (mmol Trolox g^−1^).

The free radical scavenging by ABTS radical was determined according to Yu et al. [38]. A volume of 88 μL of potassium persulfate (140 mmol/L) was added to 5 mL of ABTS (7 mmol/L). The mixture was stored in a covered bottle in the dark and at room temperature for 16 h. The ABTS solution absorbance was adjusted at 0.70 ± 0.05 at the 734 nm in spectrophotometer. Then, 188 μL of ABTS^•+^ and 12 μL of the sample or standard were placed in each well of the microplate, one well with 188 µL of ABTS and 12 µL distilled water served as blank. The mixture remained in the dark for 2 h at room temperature and the absorbance was measured at 734 nm. The results obtained were expressed in mmol of Trolox per gram (mmol Trolox g^−1^).

The ferric reducing antioxidant power assay (FRAP) was determined according to the methodology described by Yu et al. [38]. The working FRAP solution was prepared by a mixture of acetate buffer (300 mmol/L, pH 3.6), TPTZ (10 mmol/L) solubilized in HCl (40 mmol/L) and ferric chloride (20 mmol/L), in the ratio 10/1/1 (*v*/*v*/*v*), respectively. Then, 20 μL of sample, standard, or distilled water (blank) were added directly to the 96-well microplate followed by 280 μL of the working FRAP solution. The mixtures were shaken, incubated at 37 °C in the dark for 30 min, and then read at 593 nm using a microplate reader.

### 2.5. Antibacterial Activity

The antibacterial activity of spent mushroom substrate extracts from the different extraction methods was studied in different bacterial strains isolated from animals’ kidney and gastrointestinal tract (GIT): *Staphylococcus aureus* (C511 and C612; rabbit-GIT), *Enterococcus faecium* (C1 and C14; rabbit-GIT), *Aeromonas hydrophila* (C2GSPA1; fish-GIT), *Pseudomonas aeruginosa* (C3GSPR1; fish-GIT), and *Vibrio fluvialis* (RimA1TCBS; fish-kidney). These isolates belonged to MJS collection and were located at the antimicrobial, biocides, and biofilms unit department. For quality control purposes, two reference strains from the American Type Culture Collection (ATCC) were used, namely *Escherichia coli* ATCC 25922 and *S. aureus* ATCC 25923.

To evaluate the impact of polyphenolic extracts derived from spent mushroom substrate on the growth of bacterial strains, antimicrobial activity was assessed using a modified disc diffusion method, as initially described by Bauer et al. [39]. Briefly, the bacterial inoculum was obtained by introducing a colony isolated from pure strains at 0.9% NaCl solution, with the turbidity adjusted to 0.5 McFarland standard units. These inoculums were applied with a sterile cotton swab onto Petri dishes (90 mm of diameter) containing 20 mL of Mueller–Hinton agar. Additionally, two different types of sterile 6 mm paper discs were prepared. One type was impregnated with 10 μL of the polyphenolic extract (1.5 g mL^−1^), prepared in a 10% dimethyl sulfoxide (DMSO) solution, and the other using a combination of the extracts the antibiotic (gentamicin of 10 μg; CN10), in order to assess if the extracts could potentiate antibiotic activity. Subsequently, these paper discs were uniformly placed on the surface of the agar plate previously seeded with the bacterial inoculum. Further, the plates were incubated overnight at 35 ± 2 °C, and inhibitory zone diameters (in mm) around the discs with different extracts were measured. A negative control (10 μL of DMSO) and a positive control (CN10) were included. All experiments were performed in duplicate (n = 2).

The antibacterial activity evaluation was carried out using the following equation: %RIZD = ((IZD sample − IZD negative control)/IZD antibiotic standard) × 100%

% RIZD: percentage of the relative diameter of the inhibition zone, measured in mm.IZD: inhibition zone diameter measured in mm [40].

This equation compensates for the eventual inhibitory effects of any solvents distinct from water on the inhibitory zone. Additionally, the antibacterial effects of the extracts assessed were classified according to the following activity score: 0%—without effect; 0–<100%—less effective than an antibiotic; >100%—more effective than an antibiotic; Δ—extract effective and antibiotic without effect [41].

### 2.6. Statistical Analysis

The data obtained were subjected to variance analysis (ANOVA) and a multiple range test (Tukey’s test/*t* student test) for a *p* value < 0.05, using JMP statistics 7.0 software (JMP, Cary, NC, USA). All sample results were presented as mean values ± standard deviation (n = 9). Pearson correlation analysis was also performed to verify relationships between selected variables, which values were normalized to 0–100 range, considering the maximum value obtained from each assay (GrapPad Prism 10) originating a principal component analysis (PCA) and a heatmap (Past 4.03 Statistical analysis software).

## 3. Results

### 3.1. Phenolic Content

Analyzing the outcomes presented in Table 2, it becomes evident that significant differences existed among the extraction methods employed. The most promising results were consistently achieved through a long extraction of 24 h extraction (LE), conducted at high temperature of 50 °C (HT), utilizing a solution of 50% (*v*/*v*) ethanol (et) (referred to as et-LE-HT) in all phenolic composition assays. Following closely behind, the combined approach (et-COMB-HT), which entails a 24 h high-temperature extraction coupled with ultrasound extraction, exhibited remarkable performance in both total phenols and *ortho*-diphenols assays. In contrast, extractions employing distilled water as the solvent consistently yielded lower phenolic content. Furthermore, it is noteworthy that longer extraction durations at elevated temperatures (LE-HT) consistently outperformed shorter extraction methods (Appendix A).

### 3.2. Antioxidant Capacity

An in-depth analysis of the findings presented in Table 3 unveils noteworthy disparities among various extraction methods. For antioxidant capacity, also the most remarkable outcomes were consistently obtained through the 24 h extraction process at 50 °C employing a 50% (*v*/*v*) ethanol solution (et-LE-HT) in all assessments of phenolic composition. Following closely, the et-COMB-HT method, which combines the 24 h extraction with ultrasound, yielded same range values across all assessments, except for the ABTS assay, where no significant variations were observed. It is worth emphasizing that, apart from the FRAP assay, extractions conducted using distilled water as the solvent generally produced a lower capacity. Furthermore, it was observed that longer extraction processes conducted at elevated temperatures consistently outperformed shorter extraction procedures (Appendix A).

### 3.3. Antibacterial Activity

The results regarding the antibacterial activity of the spent mushroom extracts are presented in Table 4. In this context, the extracts exhibited no inhibitory effect against the majority of Gram-positive bacteria, namely *S. aureus* (C511 and C612 strains) and *E. faecium* (C1 and C14 strains). Nonetheless, some extracts inhibited the bacterial growth of *S. aureus* (ATCC 23235), achieving a reduction of 38.9% compared to the antibiotic. Regarding the studied Gram-negative isolates, the extracts alone showed antibacterial effect in all strains, except in *E. coli* (ATCC 25922). In this context, the antibacterial activity ranged from 30.4% (*V. fluvialis* RimA1TCBS) to up to 83% (*P. aureginosa* C3GSPR1) of the positive control (CN10).

Remarkably, when assessing the antibacterial effect of the combination of the extracts with the antibiotic, it was observed that some extracts potentiated the antibiotic’s effect (>100%) against *E. faecium* C1 (102.9%) and C14 (103.1–109.4%) strains, with the maximum enhancement observed in extracts obtained from et-COMB-HT, and against *P. aureginosa* C3GSPR1 (108.3%) in the long extraction method using ethanol as solvent and high temperature (et-LE-HT).

Comparing the antibacterial effect of the different extracts by extraction methods, it was observed that long and the combined extractions using high temperatures and ethanol as solvent (et-LE-HT; et-COMB-HT) had an effect against *S. aureus* (ATCC 23235), *A. hydrophila* (C2GSPA1), *P. aeruginosa* (C3GSPR1), and *V. fluvialis* (RimA1TCBS). Surprisingly, similar results were observed in the ultrasound-assisted extraction using water as solvent. Moreover, significant differences in antibacterial activity were not observed when comparing both solvents. 

### 3.4. Correlation and Principal Components Analysis

#### 3.4.1. Phenolic Content and Antioxidant Capacity

In our current study, we aimed to evaluate the correlation between phenolic compounds obtained from spent mushroom subtracts and antioxidant capacity. The observed correlations between these variables were consistently positive, signifying that higher phenolic content exerts a positive influence on antioxidant capacity. To comprehensively visualize the correlation between the phenolic content and antioxidant capacity, we employed principal component analysis (PCA) and a heatmap, illustrated in Figure 3A,B, respectively. Specifically, we found that total phenols played a substantial role in enhancing the outcomes of the DPPH and ABTS assays, while *ortho*-diphenol content had a significant impact on the ABTS and FRAP assays results. However, the flavonoid content showed limited relevance across assays, except for the ABTS assay.

Of the six variables under consideration, three are associated with phenolic content, while the remaining three are linked to measurements of antioxidant capacity (Figure 3A). The cumulative contribution of the first two principal components (PCs) explains 92.73% of the total variance within the dataset. Specifically, the first PC elucidates 84.95% of this variance, whereas the second PC elucidates 7.78%. Upon closer examination of the first PC, it becomes apparent that all variables exhibit positive loadings. Additionally, the second PC showcases positive loadings associated with high *ortho*-diphenol content. Samples positioned within the first quadrant are anticipated to boast a substantial concentration of both *ortho*-diphenols and total phenols, implying a superior performance in antioxidant capacity tests. Conversely, samples residing in the third quadrant are likely to feature lower overall phenolic content, resulting in a diminished antioxidant activity. This interpretation closely aligns with the previously presented correlation between variables, underscoring the pivotal role of phenolic content in influencing antioxidant capacity.

#### 3.4.2. Phenolic Content and Antibacterial Activity

The relationship between phenolic content and the antibacterial properties of spent mushroom substrate remains relatively unexplored. To investigate this, a PCA (principal component analysis) and a heatmap were conducted, detailed in Figure 4A,B, respectively. The heatmap analysis revealed minimal correlations between these variables, suggesting a degree of independence between the phenolic content and antibacterial activity. This suggests that the presence or absence of phenolic compounds in the samples might not strongly correlate with their antibacterial effects against the tested bacterial strains. Moreover, the calculated eigenvalues did not display a clear drop-off, indicating that the initial principal components were not significantly more informative than the subsequent ones. The absence of a distinct “elbow” in the eigenvalue plot challenges the traditional approach of selecting a small number of principal components for analysis in this scenario. Given these complexities within the dataset, alternative techniques designed to uncover non-linear relationships could be more suitable in revealing latent patterns that PCA might have missed. Techniques like t-SNE (t-distributed stochastic neighbor embedding) or UMAP (uniform manifold approximation and projection) are potential avenues worth exploring to capture these nuances. These methods could offer a more comprehensive understanding of the underlying relationships that PCA might not fully elucidate.

## 4. Discussion

### 4.1. Phenolic Content

#### 4.1.1. Total Phenols

In our current study, the total phenolic content displayed a considerable range, spanning from 85.54 to 250.92 mg of gallic acid equivalent per gram (mg GA g^−1^ extract), within the SMS extracts. It is worth noting that limited prior research exists regarding the total phenolic content in various spent mushroom substrates. Therefore, for the sake of comparison, we have drawn upon studies primarily focused on mushroom fruiting bodies and mycelia to establish reference values. Chang et al. [42] conducted an analysis of total phenols within extracts obtained from *Pleurotus ostreatus*’ spent mushroom substrate, yielding a value of 2.49 ± 0.62 mg GA^−1^ extract. Compared to our results, this value can be possibly explained due to the extreme temperature used during the 2 h extraction (95 °C). Although some reports suggest that lignin may be degraded at high temperatures, leading to a rise of phenolic acids, thermal degradation is the most common mechanism used to explain the decrease in polyphenol yield during high-temperature extractions [43]. Nonetheless, thermal degradation alone does not explain the decline in phenolic yield at temperatures above 90 °C, and all the factors, such as solvent, pressure, and methodology, must be considered. Similarly, Sułkowska-Ziaja et al. [44] examined the total phenolic compounds in five mushroom fruiting bodies, namely *Daedaleopsis confragosa*, *Fomitopsis pinicola*, *Gloeophyllum sepiarium*, *Laetiporus sulphureus*, and *Piptoporus betulinus*. The mushrooms were dried, grinded, and treated with 2M hydrochloric acid at 100 °C for 2.5 h. The total content of phenolic compounds ranged from 6.88 mg g^−1^ DW (*D. confragosa*) to 21.88 mg g^−1^ DW (*F. pinicola*). In a study by Wu et al. [45], the total phenol content was assessed in 80% (*v*/*v*) methanolic and aqueous extracts of *L. edodes* mycelia, utilizing an extraction temperature of 80 °C, with results indicating a higher content in the water extracts (6.2 mg GAE g^−1^ DW) compared to the methanol extracts (5.9 mg GAE g^−1^ DW).

Additionally, Reis et al. [46] assessed the antioxidant capacity for five different mushrooms and their mycelia, namely *A. bisporus* (white and brown), *Pleurotus ostreatus*, *Pleurotus eryngii*, and *L. edodes*. The total phenolic content varied between 7.14 (*P. eryngii*) and 37.33 (*A. bisporus*-brown) mg GAE g^−1^ extract for the fruiting body of the mushrooms, and between 4.22 (*A. bisporus*–white) and 12.53 (*L. edodes*) mg GAE g^−1^ extract for the mycelia. Moreover, Orhan et al. [47] analyzed ethanolic extracts from polyporoid fungi and reported a total phenolic content ranging from 2.50 mg GAE g^−1^ of extract (*Phellinus gilvus*) to 47.29 mg GAE g^−1^ of extract (*Fomes fomentarius*). Although the temperatures used in these studies are similar to the one presented in the present study, and the solvent is the same as the one used in the present study, other factors, such as extraction time, could possibly explain the lower yields [48]. Lavega et al. [49] evaluated the antioxidant activity of 13 commonly cultivated mushrooms in Spain. Their findings revealed a range of total phenol content, with values varying between 1.64 (*Pholiota nameko*) and 12.09 (*A. bisporus*) mg GAE g^−1^ extract for the fruiting body, 1.31 (*Pleurotus ferulae*) and 16.43 (*Hericium erinaceus*) mg GAE g^−1^ extract for the mycelia, and 5.03 (*A. bisporus*) and 14.63 (*P. nameko*) mg GAE g^−1^ extract for the spent mushroom substrate. The corresponding values for *L. edodes* were 3.27, 1.87, and 9.96, respectively. The content of phenolic compounds is higher in most of the spent substrates than in the fruiting bodies and mycelia, suggesting that during growth there’s an accumulation of phenolic products from the degradation of organic matter, lignin for example. These values are comparable to common edible mushrooms investigated by Dubost [50], 4.17 mg GAE g^−1^ DW in *Grifola frondosa*, 4.27 mg GAE g^−1^ DW in *Pleurotus ostreatus*, 4.32 mg GAE g^−1^ DW in *L. edodes*, and 10.65 mg GAE g^−1^ DW in *A. bisporus*. Similarly, another study reported a phenolic content of 1.84 ± 0.05 mg GAE g^−1^ DW for used spent mushroom [51]. This value was obtained from the grinded dried SMS (50 °C) with no further extraction.

In a different study, the total phenolic compounds in methanolic extracts of fruiting bodies from ten different inedible mushrooms were assessed. The results varied between 9.62 (*Pluteus murinus*) and 387.70 (*F. pinicola*) mg GAE g^−1^ of extract [52], while Dimitrijević et al. [53] reported values ranging between 41.90 (*Lactifluus volemus*) and 243.60 (*Butyriboletus regius*) using a methanol ultrasonic bath (25 °C; 15 min). These results are comparable to the ones obtained in the present study, although the solvents used were not the same, the temperatures used (25–35 °C) are considered safe extraction temperatures for polyphenols according to some authors [43].

In comparison to other studies, our findings are particularly noteworthy as the extracts were obtained from spent mushroom substrate, a by-product of the mushroom industry consisting of *L. edodes* mycelia and straw after two harvesting cycles.

#### 4.1.2. *Ortho*-Diphenols

In the present study, *ortho*-diphenol content varied between 214.52 and 658.19 mg GA g^−1^ extract. Not much research can be found on the *ortho*-diphenol content of different spent mushroom substrates, meaning that studies using mushrooms’ fruiting bodies and mycelia were used to compare values with the ones of the present study.

Garcia et al. [54] studied the phytochemical composition of aqueous and methanol extracts of *L. edodes* var. Koshin and Donko; the content of *ortho*-diphenols in aqueous extracts from var. Koshin (0.11 ± 0.02 mg GAE g^−1^ DW) was significantly higher than in methanolic extract (0.04 ± 0.01 mg GAE g^−1^ DW). Similarly, the content of *ortho*-diphenols in aqueous of var. Koshin extract (0.11 ± 0.02 mg GAE g^−1^ DW) was significantly higher than *ortho*-diphenols in aqueous var. Donko extract (0.08 ± 0.01 mg GAE g^−1^ DW), indicating that there were differences between the solvents used for extraction. The limited availability of literature on this subject presents a hurdle for making result comparisons. Nevertheless, within the realm of exploring plants with medicinal properties, our findings demonstrate a commendable overall efficacy.

#### 4.1.3. Flavonoids

In the present study, flavonoid content varied between 25.80 and 90.93 mg of catechin equivalent per gram (CAT g^−1^) of extract. Not much research can be found on the flavonoid content of different spent mushroom substrates, meaning that studies using mushrooms’ fruiting bodies and mycelia were used to compare values with the ones of the present study.

Palacios et al. [55] studied the total flavonoid content of eight different edible mushrooms (*A. bisporus*, *B. edulis*, *Calocybe gambosa*, *Cantharellus cibarius*, *Craterellus cornucopioides*, *Hygrophorus marzuolus*, *Lactarius deliciosus*, and *P. ostreatus*). Extracts were obtained by stirring with methanol at 65 °C for 24 h. Results varied between 0.9 (*P. ostreatus*) and 3.0 (*L. deliciosus*) mg CAT g^−1^ DW. Without additional information on the extraction yield becomes difficult to compare these results to the ones of the present study. Nevertheless, considering our extraction yield of 10%, these extracts appear to be very in line with the ones of this study.

In a similar study, results varied between 0.35 (*A. bisporus*) and 0.98 (*P. ostreatus*) mg CAT g^−1^ DW [56], when using methanol extracts acidified with 0.5% HCl. The pH of the extraction media also plays an important role in determining the effect of temperature; at lower temperatures (up to 60 °C), pH plays a significant role in anthocyanin’s thermal stability. Another study used four different solvents (petroleum ether, ethyl acetate, methanol, and water) to fractionate the soluble compounds, followed by a boiling water extraction. Results varied between 6.38 (ethyl acetate extract) and 7.79 (methanol extract) mg CAT g^−1^ extract for *P. ostreatus* [57]. Islam et al. [58] assessed the flavonoid content of 43 mushrooms commonly consumed in China. Extractions were carried out at room temperature with 5 mL extraction solvent (acetone, water, and acetic acid) for 3 h. Results ranged between 0.05 (*Auricula auricula-judae*) and 5.90 (*Chroogomphus vinicolor*) mg CAT g^−1^ sample.

Several studies were found to assess the flavonoid content but expressing it in different units, mg of quercetin equivalent per gram (QE g^−1^) DW. Since the units are not the same, these values cannot be directly compared with the ones obtained in this study. Nevertheless, these values can give us an idea of the flavonoid content. Chuang et al. [51] evaluated the antioxidant capacity of the spent mushroom substrate (referred to as waste mushroom compost) of *P. eryngii*, obtaining a flavonoid content of 1.20 ± 0.26 mg QE g^−1^ extract. Tel et al. [59] studied the effect of different solvent in extracting biological compounds from mushrooms of the *Tricholoma* species. Its content in flavonoids ranged between 1.4 (*Tricholoma terreum*; hexane extract) and 16.2 (*Tricholoma fracticum*; methanol extract) mg QE g^−1^ extract. Another study evaluated the flavonoid content of four wild edible mushrooms (*P. eryngii*, *Termitomyces robustus*, *Piccnoporrus cinnabarinus*, and *P. ostreatus*) and the results ranged between 45.94 and 66.24 mg QE g^−1^ extract.

The results of the present study are very interesting, specifically when looking at the results per gram of extract, since the lowest value obtained in this study is much higher than the highest value of the literature found.

### 4.2. Antioxidant Capacity

#### 4.2.1. ABTS

In our present study, the results of the ABTS assay exhibited a range between 0.391 and 0.906 mmol Trolox per gram of extract. It is noteworthy that there is limited research available on the antioxidant capacity of various spent mushroom substrates. As a result, we resorted to studies that primarily utilized mushroom fruiting bodies and mycelia to provide a basis for comparison.

Garcia et al. [54] evaluated the antioxidant capacity of *L. edodes* Donko and Koshin varieties. They observed higher activity in the aqueous extract of var. Koshin (1.53 ± 0.13 mmol Trolox per gram of dry weight (mmol Trolox/g DW)) compared to the methanolic extract of the same variety (0.77 ± 0.10 mmol Trolox/g DW). There was also a significant difference between the aqueous extract of var. Koshin (1.53 ± 0.13 mmol Trolox/g DW) and the aqueous extract of var. Donko (1.17 ± 0.10 mmol Trolox/g DW). Overall, the aqueous extracts exhibited higher antioxidant outcomes.

In another study assessing the antioxidant activity of mushrooms intended for consumption (*A. bisporus* and *P. ostreatus*), the antioxidant activity of fresh mushrooms ranged from 30.15 to 37.3 mmol Trolox per 100 g of DW (0.302–0.373 mmol Trolox/g DW) [56].

Similarly, when investigating the antioxidant activity of different extracts of *Lignosus tigris* mushrooms, Yap et al. [60] found that the ABTS^•+^ assay values ranged between 0.01 and 2.00 mmol Trolox per gram of extract, with the hot water extract yielding lower capacity compared to the cold-water extract and the methanol extract.

Another study examined the ABTS^•+^ free radical scavenging capacity of extracts from eight different mushroom species using three different solvents. The results varied between 1.70 and 65.98 μmol Trolox per gram of extract, which is equivalent to 0.017 and 0.660 mmol Trolox per gram of extract [61].

Overall, in the ABTS assay, our results, expressed in mmol of Trolox per gram of extract, align with those found in the literature. This underscores the potential significance of this by-product, as its antioxidant properties are comparable to those of some of the mushroom fruiting bodies studied, making it a valuable source of antioxidants.

#### 4.2.2. DPPH

In the present study, DPPH assay results varied between 0.240 and 1.449 mmol Trolox g^−1^ extract. Not much research can be found on the antioxidant capacity of different spent mushroom substrates, meaning that studies using mushrooms’ fruiting bodies were used to compare values with the ones of the present study.

In line with our findings, Yap et al. [60] concluded that the DPPH assay of extracts obtained from *Lignosus tigris* mushroom, varied between 0.18 and 2.53 mmol Trolox g^−1^ extract, both results obtained with the methanol extract. Morales et al. [62] determined the antioxidant capacity of *L. edodes*, more specifically water extracts of *L. edodes* fruiting bodies. Results ranged between 0.11 and 0.13 mmol Trolox g^−1^ extract for the DPPH assay.

Jaworska et al. [56] assessed the antioxidant capacity of prepared for consumption mushrooms (*A. bisporus* and *P. ostreatus*). When considering only the results of the fresh mushrooms, the antioxidant activity (DPPH) ranged between 18.99 and 24.01 mmol Trolox per 100 g of DW (0.190–0.240 mmol Trolox g-1 DW). Islam et al. [58] studied the phenolic profile and antioxidant activity of 43 commonly consumed mushrooms in China. When analyzing the DPPH free radical scavenging activity assay results (expressed in μmol Trolox g^−1^ DW), of the 43 mushroom samples, the higher values were recorded in *B. aereus* (18.56), *Boletus pinophilus* (17.74), and wild *Boletus aereus* (17.58).

In the context of the DPPH assay, our study’s results consistently align with the broader body of literature, reinforcing the observations made in the previous assay. This consistency suggests a degree of reliability in the antioxidant activity measurements of the extract under investigation, which may further support its potential utility in various applications.

#### 4.2.3. FRAP

In the present study, FRAP assay results varied between 0.434 and 1.197 mmol Trolox g^−1^ extract. Not much research can be found on the antioxidant capacity of different spent mushroom substrates, meaning that studies using mushrooms’ fruiting bodies and mycelia were used to compare values with the ones of the present study.

Lavega et al. [49] compared the antioxidant activity in the mycelia, fruiting bodies and spent mushroom substrate of 13 edible mushrooms. The ferric reducing antioxidant power assay, expressed in mmol Trolox g^−1^ extract, the values obtained for the fruiting bodies varied from 4.36 to 223.83. *A. bisporus*, *A. bisporus* var *brunnescens*, and *A. bisporus* var *subrufesnces* (101.89, 181.49, and 223.83, respectively) showed significantly higher values than the other species. While the lowest FRAP value was observed from the extracts of *P. ostreatus*, *P. nameko*, and *F. veluptites* var white (4.36, 9.88, and 10.94, respectively). Sulkowaska-Ziaja et al. [44] assessed the antioxidant activity in some species of mushroom from Poland. Results varied between 3.53 and 87.82 mmol Trolox kg^−1^ DW, or between 0.003 and 0.088 mmol Trolox g^−1^ DW.

This assay is the first where the literature showed very different results, with the ones of the present study being clearly outperformed by the ones of Lavega et al. [49]. Nevertheless, when looking at the results expressed per gram of DW, our values are aligning with the results found in the literature but overall are not as relevant as the previous results (ABTS and DPPH).

### 4.3. Antibacterial Activity

Concerning the potential role of spent mushroom substrate as a natural antibacterial compound, the present study revealed that most extracts exhibited no significant effect on the tested isolates when administered individually. In contrast, certain extracts reduced the antibiotic activity used as the positive control (CN10). Conversely, extracts subjected to high temperatures exhibited a moderate antibacterial effect against specific isolates among those tested.

Interestingly, Zhu et al. [63] isolated a water-soluble polysaccharide from the spent mushroom substrate of *L. edodes* and studied its antibacterial activity against *E. coli* and *S. aureus*, as well as *Sarcina lutea* (known as *Microccocus lutea*), which was not considered in the present study. Their results showed that the extracts with a higher concentration (67.25%) of Rhamnose exhibited a higher antibacterial effect. The minimal inhibitory concentration of this extract against *E. coli*, *S. aureus*, and *S. lutea* were 12.5, 25, and 100 μg/mL, respectively. In fact, this study verified a positive correlation between the polysaccharide content in SMS extracts and antibacterial activity

As far as we know the Asri et al. [64] study is the only report regarding the antibacterial activity of the *L. edodes* mushrooms ethanolic extracts. At a concentration of 1.0 g mL^−1^, it was observed that the extracts exhibited an antibacterial effect against all bacterial strains tested, namely *S. aureus* and *E. coli*. This contrasts with the present study, even though the same methodology was employed. A different study addressed the antibacterial activity and synergetic effect of *L. edodes*, var Koshin, var Donko, aqueous, and methanolic extracts [54]. In their study, a concentration of 1.0 g mL^−1^ and both varieties of methanolic extracts were not effective against either isolate tested, including the Gram-negative isolates were resistant to all extracts tested. These results were obtained for mushroom extracts (1.0 g mL^−1^), and even though some of the extracts of the present study were “moderately effective” against some isolates and mostly “noneffective” against the other isolates, the results are still interesting when considering that the extracts are from spent mushroom substrate after two production cycles at a similar/comparable concentration (1.5 g mL^−1^). Moreover, despite methodology differences, the potential of shiitake extracts has been reported to significantly reduce the growth of bacterial isolates, including pathogenic bacteria [65,66,67,68].

As one of the goals of this study was to assess the viability of using spent mushroom substrate as a prebiotic in animal diet, this assay hints at its possibility. However, more research should be envisaged on this matter. In the context of SMS antibacterial activity, the scarcity of relevant literature hinders direct comparison with these findings, as this by-product has yet to be previously analyzed for antimicrobial activity. Furthermore, as an initial exploration, this study can open doors to inventive uses of these by-products as a natural alternative approach in addressing clinically significant multidrug-resistant bacteria, thus diminishing the dependence on traditional antibiotics. The significant rise in bacterial resistance rates has significantly compromised the efficacy of existing antibiotics. Considering this global public health challenge, researching potential natural alternative compounds and complementary therapies has become paramount in addressing the mounting threat of antibiotic-resistant bacteria [69]. This imperative approach ensures a sustainable trajectory, including human healthcare and disease management [70].

One of the main limitations of this preliminary study was the low number of bacterial isolates tested. There is a need for a more in-depth exploration of the specific bioactive components present in SMS, elucidating their role in conferring antibacterial properties and the associated underlying mechanisms. Additionally, our work focused on pathogenic bacteria, overlooking the potential impact on beneficial bacteria. Future research should address this gap, particularly considering the potential application of extracts in animal testing. The repercussions on the gut microbiota remain unexplored, necessitating a thorough investigation to discern any unforeseen effects. Hence, it should be essential to identify specific phenolic compounds present in SMS and their characteristics and bioavailability within the gastrointestinal tract.

Additionally, exploring innovative methodologies, such as advanced analytical techniques and in vitro gut models, will enhance the precision of studying phenolic compound bioavailability. This multifaceted approach will contribute to a more comprehensive understanding of the potential health-promoting effects of consuming SMS-derived phenolic compounds. Regarding the limitations of this study, it is essential to consider the practical aspects of incorporating SMS extracts into large-scale production processes. Moreover, collecting samples periodically throughout the extraction process is advisable to pinpoint the optimal extraction time, mitigating costs associated with prolonged extraction durations.

## 5. Conclusions

In light of escalating sustainability concerns and waste management, repurposing spent mushroom substrate (SMS) emerges as a critical need. This initial study underscores its robust antioxidant capacity, strongly linked to a significant presence of phenolic compounds. Additionally, SMS shows promising antibacterial properties, marking it as a valuable by-product for diverse applications in both animal and human domains.

However, further research is essential to fully integrate SMS across several sectors, especially as a natural alternative to combat the surge in multi-resistant bacteria. The study’s findings underscore the potential for innovative uses of by-products, addressing pivotal challenges in healthcare and sustainability.

These discoveries resonate with principles of the circular economy and industrial symbiosis, signaling SMS’s versatility in industries spanning food, cosmetics, and pharmaceuticals. Nevertheless, the commercial feasibility of SMS extraction relies on regional factors, energy usage, and solvent requirements. Future research should delve deeper into mushroom species, substrate composition, optimized extraction techniques, production cycles, and economic viability.

In essence, this research illuminates the nutraceutical qualities of *L. edodes*’ spent mushroom substrate, showcasing its adaptability for multifaceted applications. Leveraging SMS advantages can profoundly contribute to a more sustainable, circular approach within the mushroom industry, impacting both animal and human health holistically. This approach adopts a “from farm to fork” perspective while prioritizing environmental considerations.

## Figures and Tables

**Figure 1 jof-09-01200-f001:**
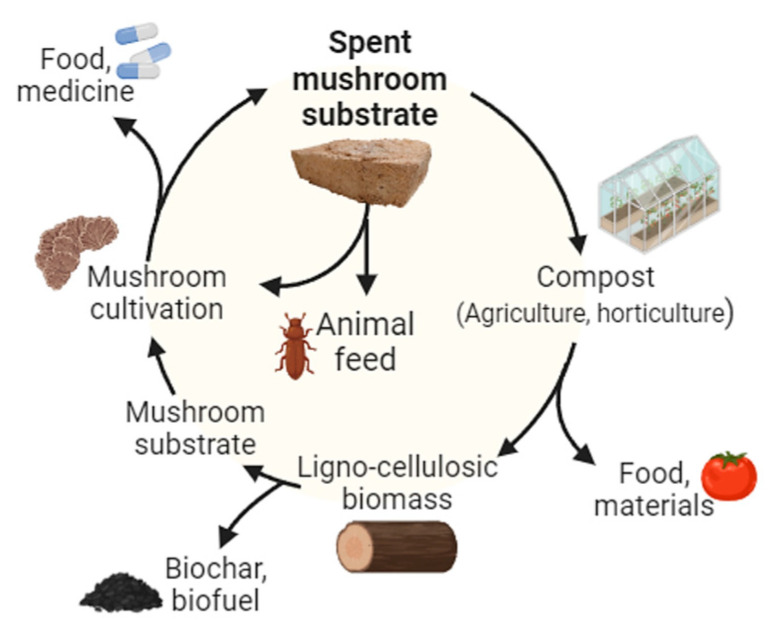
Use of spent mushroom substrate (SMS) in a circular economy context.

**Figure 2 jof-09-01200-f002:**
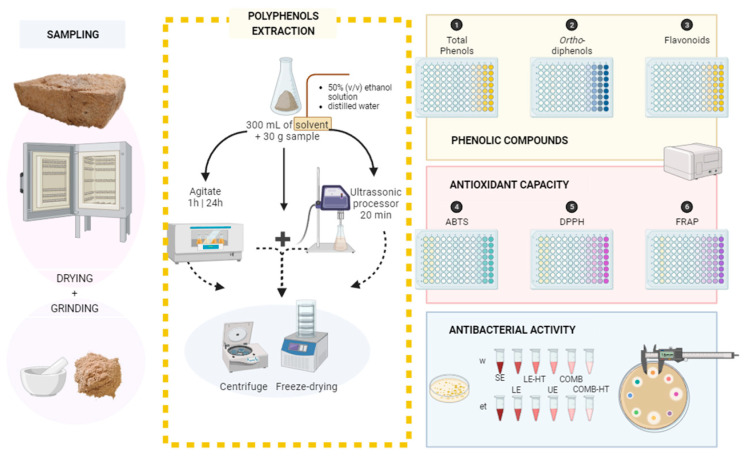
Schematic representation of the methodology used in this study. ABTS—2,2′-azino-bis(3-ethylbenzothiazoline-6-sulfonic acid) diammonium salt; DPPH—2,2-Diphenyl-1-picrylhydrazyl; FRAP—ferric-reducing antioxidant power; w—distilled water; et—ethanol 50% (*v*/*v*) solvent; SE—short extraction; LE—24 h extraction; LE-HT—24 h extraction with high temperature; UE—ultrasound extraction; COMB—combined methods (LE + UE); COMB-HT—combined methods (LE-HT + UE).

**Figure 3 jof-09-01200-f003:**
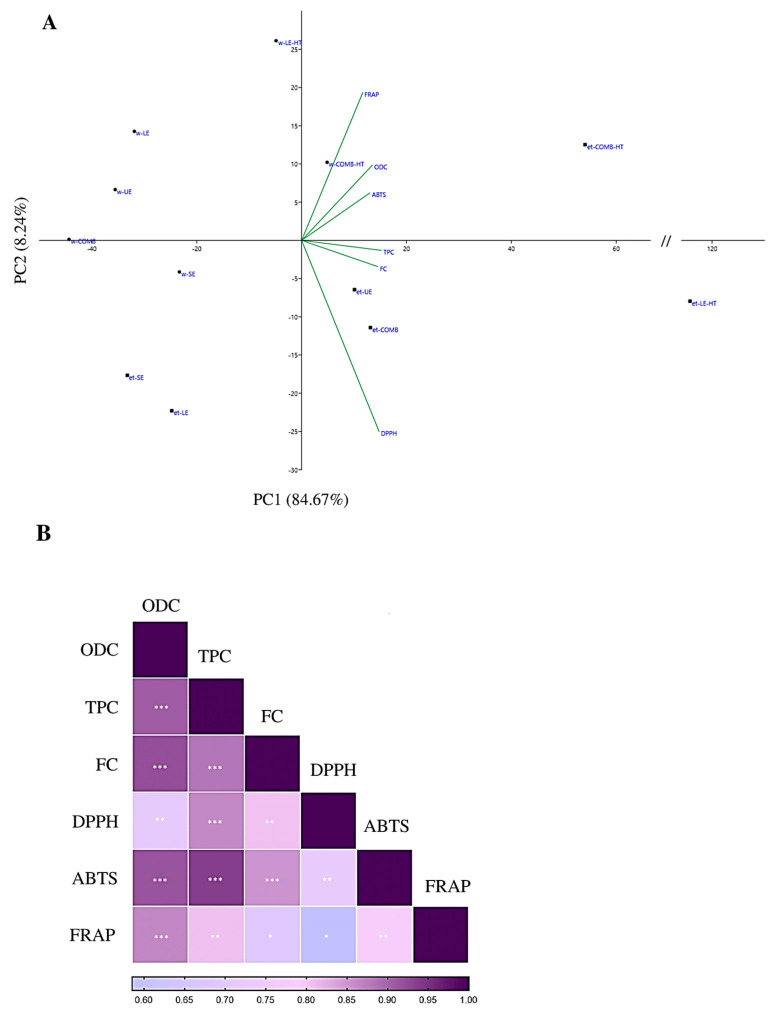
Correlation analysis of the variables phenolic content and antioxidant capacity. (**A**) PCA, (**B**) heatmap. w—distilled water; et—ethanol 50% (*v*/*v*) solvent; SE—short extraction; LE—24 h extraction; LE-HT—24 h extraction with high temperature; UE—ultrasound extraction; COMB—combined methods (LE + UE); COMB-HT—combined methods (LE-HT + UE). ABTS—2,2′-azino-bis(3-ethylbenzothiazoline-6-sulfonic acid) diammonium salt; DPPH—2,2-Diphenyl-1-picrylhydrazyl; FRAP—ferric-reducing antioxidant power; TPC—total phenols; ODC—*ortho*-diphenols; FC—flavonoids. ***, *p*-value < 0.001; **, *p*-value < 0.01; *, *p*-value < 0.05.

**Figure 4 jof-09-01200-f004:**
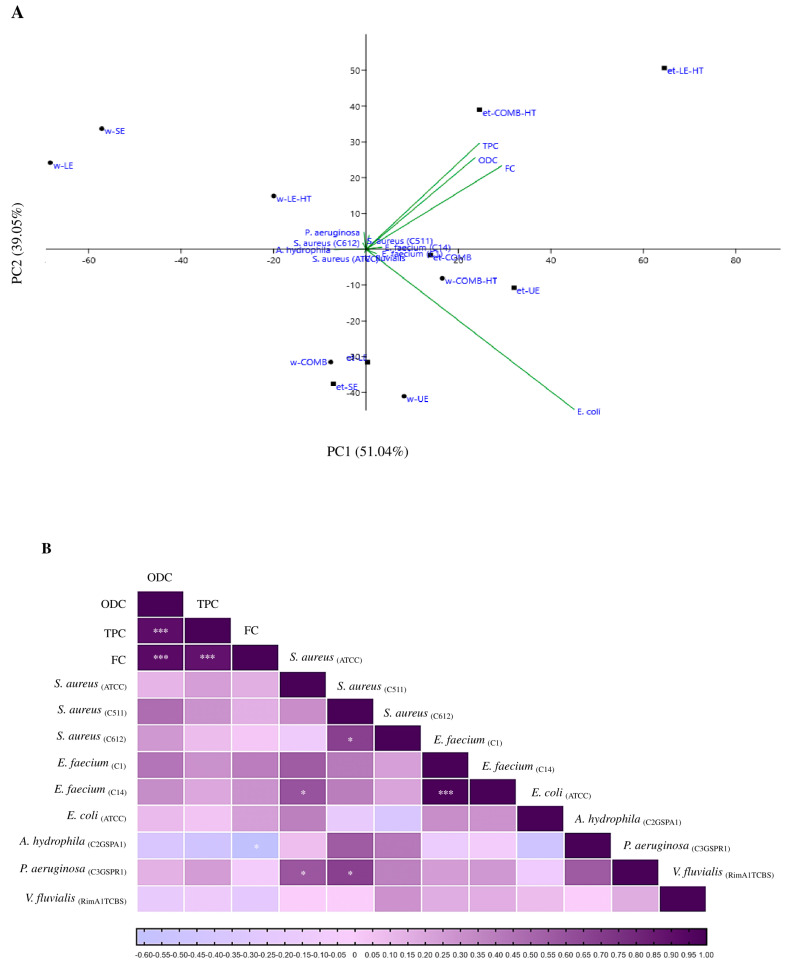
Correlation analysis of the variables phenolic content and antibacterial activity. (**A**) PCA, (**B**) heatmap. w—distilled water; et—ethanol 50% (*v/v*) solvent; SE—short extraction; LE—24 h extraction; LE-HT—24 h extraction with high temperature; UE—ultrasound extraction; COMB—combined methods (LE + UE); COMB-HT—combined methods (LE-HT + UE). ABTS—2,2′-azino-bis(3-ethylbenzothiazoline-6-sulfonic acid) diammonium salt; DPPH—2,2-Diphenyl-1-picrylhydrazyl; FRAP—ferric reducing antioxidant power; TPC—total phenols; ODC—*ortho*-diphenols; FC—flavonoids. ***, *p*-value < 0.001; *, *p*-value < 0.05.

**Table 1 jof-09-01200-t001:** Extraction methods overview.

Method	Solvent		Tmax (°C)	Ref.
SE	w	Distilled water	50	[36]
	et	50% (*v*/*v*) ethanol		
LE	w	Distilled water	40	[36]
	et	50% (*v*/*v*) ethanol		
LE-HT	w	Distilled water	50	[36]
	et	50% (*v*/*v*) ethanol		
UE	w	Distilled water	40	[37]
	et	50% (*v*/*v*) ethanol		
COMB	w	Distilled water	40	[36,37]
	et	50% (*v*/*v*) ethanol		
COMB-HT	w	Distilled water	50	[36,37]
	et	50% (*v*/*v*) ethanol		

w—distilled water; et—ethanol 50% (*v*/*v*) solvent; SE—short extraction; LE—24 h extraction; LE-HT—24 h extraction with high temperature; UE—ultrasound extraction; COMB—combined methods (LE + UE); COMB-HT—combined methods (LE-HT + UE).

**Table 2 jof-09-01200-t002:** Results of the evaluation of the composition of polyphenols in spent mushroom substrate extracts.

Samples	Total Phenols (mg GA g^−1^)	*Ortho*-Diphenols (mg GA g^−1^)	Flavonoids (mg CAT g^−1^)
w-SE	108.22 ± 3.88 ^d^	271.09 ± 9.61 ^ef^	43.40 ± 5.86 ^c^
et-SE	97.99 ± 9.60 ^de^	214.52 ± 16.68 ^g^	29.93 ± 2.23 ^de^
w-LE	88.17 ± 7.05 ^e^	282.22 ± 8.49 ^e^	25.80 ± 3.56 ^e^
et-LE	108.86 ± 10.26 ^d^	239.56 ± 3.13 ^fg^	35.70 ± 0.98 ^cde^
w-LE-HT	130.30 ± 11.99 ^c^	375.95 ± 15.27 ^cd^	38.91 ± 0.52 ^cd^
et-LE-HT	250.92 ± 13.00 ^a^	658.19 ± 15.11 ^a^	90.93 ± 10.82 ^a^
w-UE	85.54 ± 9.33 ^e^	294.77 ± 6.37 ^e^	36.47 ± 4.16 ^cd^
et-UE	137.21 ± 11.17 ^c^	399.57 ± 15.65 ^c^	62.80 ± 6.65 ^b^
w-COMB	63.91 ± 1.83 ^f^	298.02 ± 6.05 ^e^	36.89 ± 4.17 ^cd^
et-COMB	143.13 ± 5.35 ^c^	348.30 ± 25.22 ^d^	58.20 ± 2.70 ^b^
w-COMB-HT	109.46 ± 2.31 ^d^	397.53 ± 32.94 ^c^	59.72 ± 0.91 ^b^
et-COMB-HT	203.81 ± 8.73 ^b^	514.30 ± 2.54 ^b^	67.62 ± 2.49 ^b^

w—distilled water; et—ethanol 50% (*v*/*v*) solvent; SE—short extraction; LE—24 h extraction; LE-HT—24 h extraction with high temperature; UE—ultrasound extraction; COMB—combined methods (LE + UE); COMB-HT—combined methods (LE-HT + UE). The presented values are means ± standard deviation of triplicate measurements. Means in the same column with different letters were significantly different (*p* < 0.05, ANOVA, Tukey-HSD).

**Table 3 jof-09-01200-t003:** Results of the evaluation of the antioxidant capacity of the spent mushroom substrate extracts.

Samples	ABTS^•+^ (mmol Trolox g^−1^)	DPPH^•^ (mmol Trolox g^−1^)	FRAP (mmol Trolox g^−1^)
w-SE	0.391 ± 0.033 ^g^	0.458 ± 0.036 ^e^	0.638 ± 0.049 ^de^
et-SE	0.428 ± 0.016 ^fg^	0.592 ± 0.028 ^d^	0.500 ± 0.044 ^g^
w-LE	0.533 ± 0.056 ^de^	0.276 ± 0.011 ^f^	0.711 ± 0.055 ^d^
et-LE	0.486 ± 0.034 ^ef^	0.650 ± 0.041 ^c^	0.434 ± 0.021 ^g^
w-LE-HT	0.619 ± 0.019 ^bc^	0.261 ± 0.007 ^f^	0.866 ± 0.059 ^bc^
et-LE-HT	0.906 ± 0.030 ^a^	1.449 ± 0.028 ^a^	1.197 ± 0.036 ^a^
w-UE	0.408 ± 0.016 ^g^	0.259 ± 0.025 ^f^	0.600 ± 0.013 ^ef^
et-UE	0.658 ± 0.072 ^b^	0.563 ± 0.012 ^d^	0.525 ± 0.046 ^fg^
w-COMB	0.407 ± 0.030 ^g^	0.240 ± 0.011 ^f^	0.434 ± 0.033 ^g^
et-COMB	0.670 ± 0.047 ^b^	0.709 ± 0.016 ^b^	0.609 ± 0.057 ^ef^
w-COMB-HT	0.566 ± 0.056 c^d^	0.472 ± 0.047 ^e^	0.802 ± 0.020 ^c^
et-COMB-HT	0.841 ± 0.060 ^a^	0.696 ± 0.006 ^bc^	0.913 ± 0.085 ^b^

w—distilled water; et—ethanol 50% (*v*/*v*) solvent; SE—short extraction; LE—24 h extraction; LE-HT—24 h extraction with high temperature; UE—ultrasound extraction; COMB—combined methods (LE + UE); COMB-HT—combined methods (LE-HT + UE). The presented values are means ± standard deviation of triplicate measurements. Means in the same column with different letters were significantly different (*p* < 0.05, ANOVA, Tukey-HSD).

**Table 4 jof-09-01200-t004:** Antibacterial assay relative inhibition zone diameter (RIZD) results (%).

			RIZD (%)															
	Gram+										Gram−							
	*S. aureus* (ATCC 23235)		*S. aureus* (C511)		*S. aureus* (C612)		*E. faecium* (C1)		*E. faecium* (C14)		*E. coli* (ATCC 25922)		*A. hydrophila* (C2GSPA1)		*P. aeruginosa* (C3GSPR1)		*V. fluvialis* (RimA1TCBS)	
Extracts	SMS	S + CN	SMS	S + CN	SMS	S + CN	SMS	S + CN	SMS	S + CN	SMS	S + CN	SMS	S + CN	SMS	S + CN	SMS	S + CN
w-SE	0	88.9	0	86.7	0	97.1	0	88.2	0	93.8	0	0	44.4	88.9	66.7	75.0	30.4	89.1
et-SE	38.9	94.4	0	86.7	0	91.2	0	88.2	0	93.8	0	87.5	0	88.9	75.0	75.0	0	87.0
w-LE	0	88.9	0	93.3	0	97.1	0	94.1	0	100.0	0	0	50.0	94.4	75.0	83.3	30.4	87.0
et-LE	38.9	94.4	0	86.7	0	97.1	0	97.1	0	103.1	0	87.5	44.4	83.3	66.7	75.0	0	91.3
w-LE-HT	0	94.4	0	100.0	0	100.0	0	94.1	0	100.0	0	43.8	50.0	100.0	75.0	83.3	0	87.0
et-LE-HT	38.9	88.9	0	93.3	0	97.1	0	94.1	0	100.0	0	75.0	55.6	77.8	83.3	108.3	34.8	87.0
w-UE	38.9	94.4	0	93.3	0	97.1	0	97.1	0	100.0	0	100.0	44.4	88.9	66.7	75.0	30.4	87.0
et-UE	0	94.4	0	86.7	0	94.1	0	100.0	0	106.3	0	93.8	0	83.3	75.0	75.0	0	87.0
w-COMB	0	88.9	0	93.3	0	97.1	0	97.1	0	103.1	0	81.3	38.9	88.9	58.3	75.0	0	82.6
et-COMB	0	94.4	0	86.7	0	94.1	0	94.1	0	100.0	0	75.0	0	83.3	66.7	75.0	0	87.0
w-COMB-HT	0	94.4	0	93.3	0	97.1	0	97.1	0	103.1	0	81.3	44.4	88.9	66.7	75.0	30.4	87.0
et-COMB-HT	38.9	100.0	0	100.0	0	97.1	0	102.9	0	109.4	0	56.3	38.9	88.9	66.7	100.0	66.7	75.0

SMS—spent mushroom substrate; S + CN—extract combined with gentamicin of 10 μg; w—distilled water; et—ethanol 50% (*v*/*v*) solvent; SE—short extraction; LE—24 h extraction; LE-HT—24 h extraction with high temperature; UE—ultrasound extraction; COMB—combined methods (LE + UE); COMB-HT—combined methods (LE-HT + UE). The results presented are the mean of replicas (n = 2). The values under SMS represent the discs with mushroom substrate extract. The antibacterial effects of the extracts assessed were classified according to the following activity score: 0%—without effect; 0–<100%—less effective than an antibiotic; >100%—more effective than an antibiotic [41].

## Data Availability

Data are contained within the article and Appendix A.

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
