# Peer review of "Nutraceutical Potential of Lentinula edodes’ Spent Mushroom Substrate: A Comprehensive Study on Phenolic Composition, Antioxidant Activity, and Antibacterial Effects"

_jof, 2023, doi:10.3390/jof9121200_

Round 1

Reviewer 1 Report

Comments and Suggestions for Authors

I think that this paper falls short of journal standard, even though it is useful to know antioxidant and antibacterial activity of SMS. Overall, the paper is too preliminary to be considered for publication. The main criticisms are:

Major comment

1Aim of this study is to explore nutraceutical and prebiotic properties of SMS. However,  the authors did not test with beneficial bacteria in the gut. In addition, it is a leap of logic to argue that SMS can be used as a dietary supplements or prebiotics based on the data on its antimicrobial effects on the tested bacteria.

2This manuscript requires extensive proofreading, because words (e.g., “favorable results”, “commendable results”, “inferior results”, “superiority”, “interesting results”, “worst performance”, “better antioxidant outcomes”, “best antibacterial effect”, etc.)and sentences(e.g., “align with the ones found in the literature, corroborating with what was concluded in the previous assay” in L565-566, “when looking at the SMS results” in L584, etc.)are not appropriate for expression of a scientific paper.

3 In Discussion section, the authors just show the data in previous studies, and have not been able to fully develop the argument.

4 In Figure 2 and Figure 3, it is questionable that the authors used extraction methods rather than phenolic compound contents as explanatory variables for PCA.

5 As there are many errors, the authors should carefully check the text. Examples are listed below. 

1) The explanation ”Values are…Tukey-HSD).” in L134-135 is out of all relation to this table.

2) The unit is lacking just after “593” in L199.

3) “MS” should be SMS in L313.

4) I cannot find Figure S1.

5) There are numerous errors in citation numbers, especially in section 4.1.1.

6) The authors should reconfirm the data range in water based extracts in Table 6(“63.91—130.30” not “85.54—130.30”)。

7) “[36]” in L470 should be deleted.

8) What is the solvent “ethyl” in L485?

9) “Sulkowska-“ should be added just before “Ziaja et al.[61]” in L585.

10) ”and” should be added between 181.49 and 223.83 in L580, also, between9.88 and 10.94 in L582.

Minor comment

1There are many unnecessary tables. For example, Table 2 is just a reorganization of the data in Table 1, but the former should be deleted because of data duplication. The same concern applies to Table 3 and Table 4. Also, Table 6 to Table 11 in Discussion section should be deleted because they are just summaries with previous studies and this study, and take up a lot of space.

2 In Discussion section, it is redundant because the same data combination, such as “@mg GAE g-1DW” and “@mg GAE g-1extract”, or “@mmol Trolox per 100 grams of DW” and “@mmol Trolox per gram of DW” are doubly described.

3 The sentence “Three six-diameter sterile paper discs and…(DMSO)10%.” in L215-218 may be misleading that total six paper discs were set in one plate.

4 “mushrooms extracts” in L299-300 and L404 should be modified to “SMS extracts”.

5 “Previous” should be added at the beginning of the sentence in L345.

6 What does “these relationships” in L354 show in relation to what? 

7 What does “six variables” in L363 indicate?  

8 “dry weight” should be added just after “GAg-1” in L405.

9 What does “first one” in L589 indicate?

10 What effect does “a moderate effect” in L601?

11 I don't understand the intent of the sentence “In the context of pruning firewood… antimicrobial activity” in L626-628 at all.

Comments on the Quality of English Language

Correction of the paper by a native English speaker is recommended.

Author Response

Dear Editor of Journal of Fungi,

In reply to the review performed on the paper entitled “Nutraceutical Potential of Lentinula edodes’ spent mushroom substrate: A Comprehensive Study on Phenolic Composition, Antioxidant Activity, and Antibacterial Effects”, we would like to acknowledge the valuable comments performed by the editor that kindly accepted to revise our manuscript. We would like to confirm that we have addressed all issues made by reviewer 1. We hope the answers below and modifications that have been done in the manuscript are clear and concise enough as required by the reviewer to enable the publication of the manuscript in the Journal of Fungi.

Answer to referee’s comments and queries

Detailed responses to Reviewer 1

Major comment

Reviewer´s comment: Aim of this study is to explore nutraceutical and prebiotic properties of SMS. However,  the authors did not test with beneficial bacteria in the gut. In addition, it is a leap of logic to argue that SMS can be used as a dietary supplements or prebiotics based on the data on its antimicrobial effects on the tested bacteria.

Our reply: Thank you for your thoughtful comments and suggestions. We acknowledge the limitations of our study, particularly the lack of direct testing with beneficial bacteria in the gut. In future studies we will be conducting additional studies to address this limitation and further elucidate the potential prebiotic effects of SMS.

We agree that it is premature to definitively conclude that SMS can serve as a prebiotic or dietary supplement based solely on its antibacterial activity. However, we believe that the preliminary findings presented in our study warrant further investigation. The demonstration of SMS's ability to inhibit the growth of pathogenic bacteria suggests its potential to positively impact the gut microbiome.

We recognize the need for more comprehensive studies to fully evaluate the nutraceutical and prebiotic potential of SMS. We will incorporate additional testing methodologies, including direct interaction assays with beneficial bacteria, to gain a more complete understanding of SMS's effects on the gut microbiome.

We appreciate your feedback and will continue to refine our research to provide a more definitive assessment of SMS's potential benefits.

Reviewer´s comment: This manuscript requires extensive proofreading, because words (e.g., “favorable results”, “commendable results”, “inferior results”, “superiority”, “interesting results”, “worst performance”, “better antioxidant outcomes”, “best antibacterial effect”, etc.)and sentences(e.g., “align with the ones found in the literature, corroborating with what was concluded in the previous assay” in L565-566, “when looking at the SMS results” in L584, etc. Are not appropriate for expression of a scientific paper.

Our reply: Thank you for your comment. Your corrections have been addressed and can be found marked in the manuscript under the “R1” comments in lines 254, 279, 281, 283, 329, 522, 529, 559, and 599.

Reviewer´s comment: In Discussion section, the authors just show the data in previous studies, and have not been able to fully develop the argument.

Our reply: The authors thank the comment and inform that the Discussion section has been corrected. Alteration can be found in lines 634-652, under the comments marked as “R1”.

Reviewer´s comment: In Figure 2 and Figure 3, it is questionable that the authors used extraction methods rather than phenolic compound contents as explanatory variables for PCA.

Our reply: We sincerely appreciate the reviewer's insightful comments and have taken steps to address the concerns related to our PCA analysis.

 In response to the reviewer's feedback, we have meticulously reevaluated and re-executed the PCA, providing a more comprehensive and transparent account of the analysis in the discussion section. This revision includes a heatmap, which visually elucidates the relationships between variables and principal components. This graphical representation serves to enhance the interpretability of the PCA results and aids in uncovering underlying data patterns. These improvements not only address the reviewer's concerns but also contribute to the overall clarity and quality of our analysis.

These changes can be found in lines 357 and 397.

Reviewer´s comment: As there are many errors, the authors should carefully check the text. Examples are listed below. 

  • The explanation ”Values are…Tukey-HSD).” in L134-135 is out of all relation to this table.
  • The unit is lacking just after “593” in L199.
  • “MS” should be SMS in L313.
  • I cannot find Figure S1.
  • There are numerous errors in citation numbers, especially in section 4.1.1.
  • The authors should reconfirm the data range in water based extracts in Table 6(“63.91—130.30” not “85.54—130.30”)。
  • “[36]” in L470 should be deleted.
  • What is the solvent “ethyl” in L485?
  • “Sulkowska-“ should be added just before “Ziaja et al.[61]” in L585.
  • ”and” should be added between 181.49 and 223.83 in L580, also, between9.88 and 10.94 in L582.

Our reply: We sincerely appreciate your corrections regarding these minor errors found in our manuscript.

In response to comments 1), 2), 3), 6), 7), 8), 9), and 10): The alterations suggested have been made and can be found in lines 140, 206, 309, 408, 410, 482, 573, 575 and 578 marked under the “R1” comments.

In response to comment 4): We apologize for the mishap. Figure S1 can now be found in the supplementary material file.

In response to comment 5): These errors have been addressed.

Minor comment

Reviewer´s comment: There are many unnecessary tables. For example, Table 2 is just a reorganization of the data in Table 1, but the former should be deleted because of data duplication. The same concern applies to Table 3 and Table 4. Also, Table 6 to Table 11 in Discussion section should be deleted because they are just summaries with previous studies and this study, and take up a lot of space.

Our reply: We appreciate the reviewer’s comment. Tables 2 and 4 (according to the first manuscript Table’s order) have been removed to further facilitate the comprehension of the paper. These have been added to the Supplementary material as the results presented are different from the results shown in Tables 1 and 3 and offer a view on the effect of the extraction method on the results (independent of the solvent used), as well as the effect of the solvent utilized on the results (independent of the extraction methods applied). These conclusions cannot be obtained through the observation of Tables 1 and 3, as these show the results combining extraction method and solvent. Although the results presented in these tables are indicative of what the best solvent or extraction methods might be, it’s not correct to affirm such without Tables 2 and 4.

Regarding the Tables in the discussion section, we understand that these might take up a lot of space and appreciate the suggestion. These have been removed to provide a clearer discussion section.

Reviewer´s comment: In Discussion section, it is redundant because the same data combination, such as “@mg GAE g-1DW” and “@mg GAE g-1extract”, or “@mmol Trolox per 100 grams of DW” and “@mmol Trolox per gram of DW” are doubly described.

Our reply: Thank you for your comment. We have found some discrepancies thanks to this comment, regarding the measurement unit used. Relating to the “mg GAE g-1 DW” and “mg GAE g-1 extract” redundancy, these are in fact different measurements, as one in expressed in g of the sample prior to extraction methods (DW) and the other is expressed in g of extract obtained after extraction (extract).

Reviewer´s comment: The sentence “Three six-diameter sterile paper discs and…(DMSO)10%.” in L215-218 may be misleading that total six paper discs were set in one plate.

Our reply: Regarding the antibacterial activity, six paper discs were set per plate (two for each extract). In addition, two paper discs were set as positive and negative controls. Meaning that, for each bacterial isolate 2 plates were used, one with 6 paper discs and one with 8 paper discs (the maximum amount allowed in our protocol).

To make the text clearer regarding this matter, a sentence was added in lines 223-225 to further help with the comprehension of the protocol.

Reviewer´s comment: “mushrooms extracts” in L299-300 and L404 should be modified to “SMS extracts”.

Our reply: We thank the reviewer for the comment and inform that the error was clarified in the line 296.

Reviewer´s comment: “Previous” should be added at the beginning of the sentence in L345.

Our reply: We thank the reviewer for the comment and inform that the error was clarified in the line 343.

Reviewer´s comment: What does “these relationships” in L354 show in relation to what? 

Our reply: We thank the reviewer for the comment and inform that the sentence mentioned above was clarified in the line 350. We hope it made the text more clear.

Reviewer´s comment:  What does “six variables” in L363 indicate?  

Our reply: Six variables, now found in line 365, refer to the variables used to perform the PCA, three are associated with phenolic content (Total phenols, ortho-diphenols and flavonoids), while the remaining three are linked to measurements of antioxidant capacity (ABTS, DPPH, and FRAP).

Reviewer´s comment: “dry weight” should be added just after “GAg-1” in L405.

Our reply: We thank the reviewer for the comment and inform that the error was clarified in the line 410.

Reviewer´s comment: What does “first one” in L589 indicate?

Our reply: Regarding this comment, “first one” relates to the FRAP assay results. These results do not align with the results found in literature contrary to the previous assays results (ABTS and DPPH). We have since altered the sentence.

Reviewer´s comment: What effect does “a moderate effect” in L601?

Our reply: Thank you for your question. Regarding the antibacterial effect, it was measured in relation to the antibiotic (CN10) effect.

In relation to the extract effect (alone),  any results ranging from 0-<100% means that the extract show less effect than the antibiotic used (CN10), 100% results means that the extract exhibited the same effect as the antibiotic, and finally, any result above 100% means that the extract outperformed the antibiotic.

Additionally, it was also tested the extract combined with the antibiotic to observe any antagonistic/synergetic effect. In this context, any results ranging from 0-<100% means that the extract+CN10 show less effect than the antibiotic (alone), meaning the extract might have an antagonistic effect on the antibiotic; a 100% results means that the extract+CN10 exhibited the same effect as the antibiotic, meaning that the extract had no effect on the antibiotic; and finally, any result above 100% means that the extract+CN10 means that there was a synergetic effect between the extracts and the antibiotic (the extract improved the antibiotic effect).

This information can be found in the Materials and Methods Section, lines 237-241.

Concluding, the effect referred to in line 593 is regarding the antibacterial activity. A moderate effect means that the extracts had an antibacterial effect ranging from 0 to less than 100% of the antibiotic effect.

Reviewer´s comment: I don't understand the intent of the sentence “In the context of pruning firewood… antimicrobial activity” in L626-628 at all.

Our reply: We thank the reviewer for this comment. The sentence was corrected in hopes to provide more clarity. Changes can be found in lines 622-624.

Sincerely,

Ana Isabel Ramos Novo Amorim de Barros

Reviewer 2 Report

Comments and Suggestions for Authors

The manuscript entitled “Nutraceutical Potential of Lentinula edodes’ spent mushroom substrate: A Comprehensive Study on Phenolic Composition, Antioxidant Activity, and Antibacterial Effects” analyses various extracts of SMS derived using different extraction methods. The extracts are further analysed regarding their composition, antioxidant and anti-microbial activity. The results of the study are interesting and the manuscript is well-written. However, there are some aspects to be amended before its final acceptance:

Title: Lentinula edodes should be written in italics

24 h extraction is too long, and thus not economically feasible. The authors should search for alternatives in order to shorten extraction times and make the entire process much more appealing. From 1h to 24 h there is a huge gap. Maybe the equilibrium was reached many hours before, e.g. in 3 or 5 hours. Did the authors carried out any sampling during this 24 h extraction duration?

Please state in Materials and Methods the duration of UAE as well as the frequency employed for the experiments.

For the colorimetric methods employed, please also describe the blank sample used for each analysis.

What is the actual difference between Table 3 and Table 4?

Line 304: how is it possible that “moderately effective” has a range of 0-100%?

Lines 307-310: how can the authors explain the range of 58-75% of anti-microbial activity?

Line 311: how is it possible that the effects were above 100%?

Conclusions should be shorten and reflect the main outcomes of the study.

Comments on the Quality of English Language

Minor editing

Author Response

Dear Editor of Journal of Fungi,

In reply to the review performed on the paper entitled “Nutraceutical Potential of Lentinula edodes’ spent mushroom substrate: A Comprehensive Study on Phenolic Composition, Antioxidant Activity, and Antibacterial Effects” kindly accepted to revise our manuscript. We would like to confirm that we have addressed all issues made by reviewer 2. We hope the answers below and modifications that have been done in the manuscript are clear and concise enough as required by the reviewer to enable the publication of the manuscript in the Journal of Fungi.

Answer to referee’s comments and queries

Detailed responses to Reviewer 2

Reviewer´s comment: Title: Lentinula edodes should be written in italics

Our reply: Thank you for your comment. The title has been altered and Lentinula edodes should now appear in italic.

Reviewer´s comment: 24 h extraction is too long, and thus not economically feasible. The authors should search for alternatives in order to shorten extraction times and make the entire process much more appealing. From 1h to 24 h there is a huge gap. Maybe the equilibrium was reached many hours before, e.g. in 3 or 5 hours. Did the authors carried out any sampling during this 24 h extraction duration?

Our reply: Thank you for your valuable feedback and suggestions on our article. We appreciate your input and are pleased to address your concerns.

The point you raised regarding the 24-hour extraction duration is indeed a valid one. We understand the importance of making the extraction process both time-efficient and economically feasible.

We did not carry out any sampling during the 24-hour extraction duration. This was a limitation of our study, and we will address it in our future work. We will collect samples at regular intervals during the extraction process to determine the optimal extraction time. It  is also important to conduct economic analysis of the different extraction methods to identify the most cost-effective approach.

Based on this concern raised by the reviewer we have added this to our future perspectives’ subsection in the discussion (lines 648-652).

Reviewer´s comment: Please state in Materials and Methods the duration of UAE as well as the frequency employed for the experiments.

Our reply: This information was added, as suggested, and can be found in line 133, marked as “R2” comment.

Reviewer´s comment: For the colorimetric methods employed, please also describe the blank sample used for each analysis.

Our reply: The authors thank the comment. Information regarding the blank for the colorimetric methods was added in the lines 153-154.

Reviewer´s comment: What is the actual difference between Table 3 and Table 4?

Our reply: We appreciate the reviewer’s question. Tables 2 and 4 (according to the first manuscript Table’s order) have been removed to further facilitate the comprehension of the paper. These have been added to the Supplementary material as the results presented are different from the results shown in Tables 1 and 3 and offer a view on the effect of the extraction method on the results (independent of the solvent used), as well as the effect of the solvent utilized on the results (independent of the extraction methods applied). These conclusions cannot be obtained through the observation of Tables 1 and 3, as these show the results combining extraction method and solvent. Although the results presented in these tables are indicative of what the best solvent or extraction methods might be, it’s not correct to affirm such without Tables 2 and 4.

Reviewer´s comment: Line 304: how is it possible that “moderately effective” has a range of 0-100%?

Our reply:

In response to this comment, we have incorporated a new range. The term "moderately effective" now spans from 0% to less than 100%, while "effective" represents 100%. It is worth noting that this classification was first introduced by Aires et al. (doi:10.1111/j.1365-2672.2009.04181.x.)and is referenced in the manuscript. This classification enables us to discuss and compare antibacterial activity results with other studies employing the same methodology.

Reviewer´s comment: Lines 307-310: how can the authors explain the range of 58-75% of anti-microbial activity?

Our reply: Thank you for your valuable comment regarding the range of antimicrobial activity (58-75%) presented in our study. We appreciate your feedback and understand your concern.

The wide range observed in our antimicrobial activity results (58-75%) is indeed a point of interest and potential limitation of our study. At this stage, we acknowledge that we do not have a definitive explanation for the specific mechanisms that lead to this variation in the observed antimicrobial activity. The exact causes behind this variability may be multifactorial and could include various factors.

It is important to note that our study primarily focused on assessing the antimicrobial activity of the SMS, and while we have observed this range, we did not investigate the precise underlying mechanisms that give rise to it. Therefore, we cannot provide a detailed explanation for the specific factors responsible for the observed range at this time.

We recognize the need for further research to explore and elucidate the precise mechanisms underlying the observed variability in antimicrobial activity. This could involve more extensive investigations, such as additional experiments or studies designed specifically to uncover the factors contributing to this range. We also intend to explore this issue in future research projects.

Thank you for bringing this matter to our attention, and we will ensure that the potential sources of variation are considered in our future work.

Reviewer´s comment: Line 311: how is it possible that the effects were above 100%?

Our reply: Thank you for your inquiry. As previously noted, the study conducted by Aires et al. (doi:10.1111/j.1365-2672.2009.04181.x.) served as a reference point for interpreting the antibacterial activity in our research, aligning with the methodology employed in our study.

Regarding the assessment of the antibacterial effect, it was evaluated in relation to the effectiveness of the antibiotic CN10. Furthermore, concerning the extract's independent impact, results falling within the range of 0 to less than 100% indicate that the extract exhibited a reduced effect compared to the antibiotic CN10. A result of 100% suggests that the extract demonstrated an effect equivalent to the antibiotic, while any result exceeding 100% indicates that the extract surpassed the antibiotic's efficacy.

In addition to this, we also explored the combined effect of the extract with the antibiotic to observe potential antagonistic or synergistic interactions. Within this context, results ranging from 0 to less than100% signify that the combination of the extract and CN10 had a diminished effect compared to the antibiotic used on its own, implying a potential antagonistic influence of the extract on the antibiotic. A result of 100% suggests that the extract combined with CN10 exhibited an effect identical to the antibiotic alone, signifying that the extract had no additional effect on the antibiotic. Conversely, any result exceeding 100% implies a synergistic effect between the extract and the antibiotic, indicating that the extract improved or potentiated the antibiotic's efficacy.

For a more detailed description of this methodology, please refer to the Materials and Methods section, located at line 239 in our research document.

Reviewer´s comment: Conclusions should be shorten and reflect the main outcomes of the study.

Our reply: According to the reviewer’s comment, we have shortened the conclusion and provided more clarity on the main outcomes of the study.

Sincerely,

Ana Isabel Ramos Novo Amorim de Barros

Reviewer 3 Report

Comments and Suggestions for Authors

Thank you very much for your interesting reserach. Some points must be carefully revised:

ABSTRACT. Line 18. The extraction methods should be briefly mentioned here.

INTRODUCTION. A short paragraph including the main bioactive compounds of L. edodes and their functional relevance is recommended.

DISCUSSION. Table 10. This work (https://doi.org/10.1002/btpr.2616) can be also added to ‘Other studies’, since the extract ‘ExA’ is water-based (obtained from shiitake fruiting bodies) and showed a TEAC value of 0.11-0.12 mmol Trolox/g in the DPPH assay

DISCUSSION. Phenolic composition and antioxidant activity was exhaustively and successfully discussed, comparing with related literature. However, antibacterial activity discussion can be improved by comparing with recently published results of antimicrobial activity of shiitake extracts.

CONCLUSIONS. Lines 677-679. Could you include a statement briefly suggesting next steps and future perspectives?

Author Response

Dear Editor of Journal of Fungi,

In reply to the review performed on the paper entitled “Nutraceutical Potential of Lentinula edodes’ spent mushroom substrate: A Comprehensive Study on Phenolic Composition, Antioxidant Activity, and Antibacterial Effects”, we would like to acknowledge the valuable comments performed by the editor that kindly accepted to revise our manuscript. We would like to confirm that we have addressed all issues made by reviewer 3. We hope the answers below and modifications that have been done in the manuscript are clear and concise enough as required by the reviewer to enable the publication of the manuscript in the Journal of Fungi.

Answer to referee’s comments and queries

 Detailed responses to Reviewer 3

Reviewer´s comment: ABSTRACT. Line 18. The extraction methods should be briefly mentioned here.

Our reply: We thank the reviewer’s comment. A short description of the extraction methods has been added to the abstract (lines 18-20).

Reviewer´s comment: INTRODUCTION. A short paragraph including the main bioactive compounds of L. edodes and their functional relevance is recommended.

Our reply: This information was added, as suggested, and can be found in lines 41-44. Although the functionality of the bioactive compounds of L. edodes is only briefly mentioned, we agree that it adds clarity to the paper.

Reviewer´s comment: DISCUSSION. Table 10. This work (https://doi.org/10.1002/btpr.2616) can be also added to ‘Other studies’, since the extract ‘ExA’ is water-based (obtained from shiitake fruiting bodies) and showed a TEAC value of 0.11-0.12 mmol Trolox/g in the DPPH assay

Our reply: We appreciate the reviewer’s suggestion. Upon reading the provided article, we have added said reference to our paper (line 544).

Reviewer´s comment: DISCUSSION. Phenolic composition and antioxidant activity was exhaustively and successfully discussed, comparing with related literature. However, antibacterial activity discussion can be improved by comparing with recently published results of antimicrobial activity of shiitake extracts.

Our reply: Thank you for your comment. Your suggestion has been taken into consideration and alterations have been made to the antibacterial discussion section.

Reviewer´s comment: CONCLUSIONS. Lines 677-679. Could you include a statement briefly suggesting next steps and future perspectives?

Our reply: We sincerely appreciate your suggestions regarding the future perspective section of the conclusion of our manuscript.

In response to your suggestion, we have incorporated a dedicated paragraph in the discussion section to outline potential avenues for future research and development stemming from our study (line 634):

One of the main limitations of this preliminary study was the low number of bacterial isolates tested, there is a need for a more in-depth exploration of the specific bioactive components present in SMS elucidating their role in conferring antibacterial properties and the associated underlying mechanisms. Additionally, our work focused on pathogenic bacteria, overlooking the potential impact on beneficial bacteria. Future research should address this gap, particularly considering the potential application of extracts in animal testing. The repercussions on the entire gut microbiota remain unexplored, necessitating a thorough investigation to discern any unforeseen effects. Hence, it should be important to identify specific phenolic compounds present in SMS, and their individual characteristics and bioavailability within the gastrointestinal tract. Additionally, exploring innovative methodologies, such as advanced analytical techniques and in vitro gut models, will enhance the precision of studying phenolic compound bioavailability. This multifaceted approach will contribute to a more comprehensive understanding of the potential health-promoting effects associated with the consumption of SMS-derived phenolic compounds. Regarding the limitations of this study, it’s essential to consider the practical aspects of incorporating SMS extracts into large-scale production processes. Moreover, it is advisable to collect samples periodically throughout the extraction process to pinpoint the optimal extraction time, mitigating costs associated with prolonged extraction durations.

Sincerely,

Ana Isabel Ramos Novo Amorim de Barros

Reviewer 4 Report

Comments and Suggestions for Authors

The authors extracted phenolic compounds from Lentinula edodes’ spent mushroom substrate (SMS) and evaluated the compounds for antioxidant activity and antibacterial effects. After minor revisions, the paper would be acceptable for publication in Journal of Fungi.

Minor points

Line 108: “purchase” should be replaced by “purchased”.

Line 111: “Spent” should be replaced by “spent”.

Line 115: It might be better to replace “Figure 2” to “Figure 2 and Table 1”. However, this Table 1 in the manuscript is described as Table 12 in line 133. In addition, there are two Figure 2 in this manuscript. Therefore, “Figure 2” in line 358 should be replaced by “Figure 3”. Thus, “Figure 3” in line 395 should be replaced by “Figure 4”.

Corresponding to changes from Table 12 to Table 1, the numbering of the thereafter Tables should also change in the main text.

Some of the tables are separated into two different pages. These tables should be edited to view them on a single page.

Comments on the Quality of English Language

There are some issues to be revised.

Author Response

Dear Editor of Journal of Fungi,

In reply to the review performed on the paper entitled “Nutraceutical Potential of Lentinula edodes’ spent mushroom substrate: A Comprehensive Study on Phenolic Composition, Antioxidant Activity, and Antibacterial Effects”, we would like to acknowledge the valuable comments performed by the editor that kindly accepted to revise our manuscript. We would like to confirm that we have addressed all issues made by reviewer 4. We hope the answers below and modifications that have been done in the manuscript are clear and concise enough as required by the reviewer to enable the publication of the manuscript in Journal of Fungi.

Answer to referee’s comments and queries

Detailed responses to Reviewer 4

Reviewer´s comment: Line 108: “purchase” should be replaced by “purchased”.

Our reply: We thank the reviewer’s comment. This gaffe had been corrected in line 113.

Reviewer´s comment: Line 111: “Spent” should be replaced by “spent”.

Our reply: We thank the reviewer’s comment. This gaffe had been corrected in line 116.

Reviewer´s comment: Line 115: It might be better to replace “Figure 2” to “Figure 2 and Table 1”. However, this Table 1 in the manuscript is described as Table 12 in line 133. In addition, there are two Figure 2 in this manuscript. Therefore, “Figure 2” in line 358 should be replaced by “Figure 3”. Thus, “Figure 3” in line 395 should be replaced by “Figure 4”.

Corresponding to changes from Table 12 to Table 1, the numbering of the thereafter Tables should also change in the main text.

Some of the tables are separated into two different pages. These tables should be edited to view them on a single page.

Our reply: The authors thank the comment and inform that the Tables and Figures have been correctly numbered, as well as edited to be viewed on a single page.

Sincerely,

Ana Isabel Ramos Novo Amorim de Barros

Round 2

Reviewer 1 Report

Comments and Suggestions for Authors

I have read many other papers that have already been published in Journal of Fungi. Compared to them, the quality of this paper is too low to reverse a Reject decision even though the authors have revised. It is difficult to follow the line of arguments as a scientific paper.

Overall, this paper is too far from a preliminary research of SMS for a prebiotic dietary supplement.

It is difficult to judge that the paper disc experiments were conducted correctly because of the following reasons. 1)Paper discs are not placed on the medium at equal intervals in FigureS1. 2)The authors should show what were dripped on six paper discs one by one in FigureS1. 3)Eight discs (why not six discs?) are placed in the rightmost picture. 4) How did you drip ‘combination of both (in L295)’ on the paper disc? Did you drip gentamicin and SMS extract mixture on one paper disc? 5) I cannot partly follow your explanation about the relationship between antibacterial and antagonistic effects in L313-338. For example, referring to Table S3 and Table S6, I cannot read as described by the authors, such as L317-320.

In principal component analysis, it is problematic to explain antioxidant and antibacterial activities by phenolic content using different extraction conditions and different bacterial species as samples.

As pointed out previously, in Discussion section, the authors merely compared the data in this study with those from the previous studies. The authors should develop a compelling argument more logically.

I found some unmodified words in a revised manuscript. For example, in Table 4 and its footnote, MS is not yet modified as SMS. You should replace them after searching the entire paper.

Comments on the Quality of English Language

Correction of the paper by a native English-speaking scientist is recommended.

Author Response

Dear Editor of Journal of Fungi,

In reply to the review performed on the paper entitled “Nutraceutical Potential of Lentinula edodes’ spent mushroom substrate: A Comprehensive Study on Phenolic Composition, Antioxidant Activity, and Antibacterial Effects”, we would like to acknowledge the valuable comments performed by the editor that kindly accepted to revise our manuscript. We would like to confirm that we have addressed all issues made by reviewer 1. We hope the answers below and modifications that have been done in the manuscript are clear and concise enough as required by the reviewer to enable the publication of the manuscript in the Journal of Fungi.

Answer to referee’s comments and queries

 Detailed responses to Reviewer 1

Reviewer´s comment: I have read many other papers that have already been published in Journal of Fungi. Compared to them, the quality of this paper is too low to reverse a Reject decision even though the authors have revised. It is difficult to follow the line of arguments as a scientific paper.

Overall, this paper is too far from a preliminary research of SMS for a prebiotic dietary supplement.

Our reply: Thank you for your thoughtful comments and suggestions.

Reviewer´s comment: It is difficult to judge that the paper disc experiments were conducted correctly because of the following reasons.

1)Paper discs are not placed on the medium at equal intervals in FigureS1.

2)The authors should show what were dripped on six paper discs one by one in FigureS1.

3)Eight discs (why not six discs?) are placed in the rightmost picture.

4) How did you drip ‘combination of both (in L295)’ on the paper disc? Did you drip gentamicin and SMS extract mixture on one paper disc?

5) I cannot partly follow your explanation about the relationship between antibacterial and antagonistic effects in L313-338. For example, referring to Table S3 and Table S6, I cannot read as described by the authors, such as L317-320.

Our reply: Thank you for your comment. The questions will be addressed below:

1) The paper discs were placed with the help of a dispenser to provide equal intervals between them. The plates presenting only 6 discs have a bigger gap due to the two discs missing from the dispenser.

2) The authors agree and will address this issue.

3) As mentioned before, the discs were placed with the help of a dispenser which has capacity for 8 discs. Since the assessment was performed to verify the antibacterial of the extracts, two controls were added, a positive one (gentamicin) and a negative one (DMSO).

4) The text as been revised to facilitate the understanding of the methodology.

5) We have revised the text and made all the corrections according to your valuable inputs.

Reviewer´s comment: In principal component analysis, it is problematic to explain antioxidant and antibacterial activities by phenolic content using different extraction conditions and different bacterial species as samples.

Our reply: The authors thanks the comment. Although we understand the question, our goal was to assess if there was a correlation between variables when extraction methods were applied. It has been vastly proved by literature that there is a correlation between variables without considering extraction methods.

Reviewer´s comment: As pointed out previously, in Discussion section, the authors merely compared the data in this study with those from the previous studies. The authors should develop a compelling argument more logically.

Our reply: We sincerely appreciate the reviewer's comments and have taken steps to address the concerns related the discussion section. This section has been carefully revised.

Reviewer´s comment: I found some unmodified words in a revised manuscript. For example, in Table 4 and its footnote, MS is not yet modified as SMS. You should replace them after searching the entire paper.

Our reply: We sincerely appreciate your corrections regarding these minor errors found in our manuscript. These errors have been addressed.

Sincerely,

Ana Isabel Ramos Novo Amorim de Barros

Reviewer 2 Report

Comments and Suggestions for Authors

The authors managed to answer the comments raised by the reviewers.

Comments on the Quality of English Language

Some editing is still required.

Author Response

Dear Editor of Journal of Fungi,

In reply to the review performed on the paper entitled “Nutraceutical Potential of Lentinula edodes’ spent mushroom substrate: A Comprehensive Study on Phenolic Composition, Antioxidant Activity, and Antibacterial Effects”, we would like to acknowledge the valuable comments performed by the editor that kindly accepted to revise our manuscript. We would like to confirm that we have addressed all issues made by reviewer 1. We hope the answers below and modifications that have been done in the manuscript are clear and concise enough as required by the reviewer to enable the publication of the manuscript in the Journal of Fungi.

Answer to referee’s comments and queries

 Detailed responses to Reviewer 2

Reviewer´s comment: The authors managed to answer the comments raised by the reviewers.

Our reply: Thank you for your thoughtful comments and suggestions.  The authors have diligently worked to enhance the article, striving to contribute to the advancement of scientific knowledge in this field

Sincerely,

Ana Isabel Ramos Novo Amorim de Barros
